# Atorvastatin Modulates Bile Acid Homeostasis in Mice with Diet-Induced Nonalcoholic Steatohepatitis

**DOI:** 10.3390/ijms22126468

**Published:** 2021-06-16

**Authors:** Hana Lastuvkova, Fatemeh Alaei Faradonbeh, Jolana Schreiberova, Milos Hroch, Jaroslav Mokry, Hana Faistova, Zuzana Nova, Radomír Hyspler, Ivone Cristina Igreja Sa, Petr Nachtigal, Alzbeta Stefela, Petr Pavek, Stanislav Micuda

**Affiliations:** 1Department of Pharmacology, Faculty of Medicine in Hradec Kralove, Hradec Kralove, Charles University, Simkova 870, 500 03 Hradec Kralove, Czech Republic; LastuvkovaH@lfhk.cuni.cz (H.L.); alaeifaf@lfhk.cuni.cz (F.A.F.); CermanovaJ@lfhk.cuni.cz (J.S.); novaz@lfhk.cuni.cz (Z.N.); 2Department of Medical Biochemistry, Faculty of Medicine in Hradec Kralove, Hradec Kralove, Charles University, 500 03 Hradec Kralove, Czech Republic; HrochM@lfhk.cuni.cz; 3Department of Histology and Embryology, Faculty of Medicine in Hradec Kralove, Hradec Kralove, Charles University, 500 03 Hradec Kralove, Czech Republic; mokry@lfhk.cuni.cz; 4The Fingerland Department of Pathology, Faculty of Medicine in Hradec Kralove, Charles University, Hradec Kralove, 500 03 Hradec Kralove, Czech Republic; chiotcurieux@gmail.com; 5Institute of Clinical Biochemistry and Diagnostics, University Hospital, Hradec Kralove, 500 03 Hradec Kralove, Czech Republic; radomir.hyspler@fnhk.cz; 6Department of Biological and Medical Sciences, Faculty of Pharmacy in Hradec Kralove, Charles University, 500 03 Hradec Kralove, Czech Republic; ivone.c.sa@gmail.com (I.C.I.S.); petr.nachtigal@faf.cuni.cz (P.N.); 7Department of Pharmacology and Toxicology, Faculty of Pharmacy in Hradec Kralove, Charles University, 500 03 Hradec Kralove, Czech Republic; horvata1@faf.cuni.cz (A.S.); pavek@faf.cuni.cz (P.P.)

**Keywords:** nonalcoholic steatohepatitis, atorvastatin, bile acids, apical sodium-dependent bile acid transporter, deoxycholic acid

## Abstract

Bile acids (BA) play a significant role in the pathophysiology of nonalcoholic steatohepatitis (NASH). The present study evaluates the modulation of bile acid metabolomics by atorvastatin, a cholesterol-lowering agent commonly used to treat cardiovascular complications accompanying NASH. NASH was induced in mice by 24 weeks of consuming a high–saturated fat, high-fructose, and high-cholesterol diet (F), with atorvastatin administered orally (20 mg/kg/day) during the last three weeks. Biochemical and histological analyses confirmed the effectiveness of the F diet in inducing NASH. Untreated NASH animals had significantly reduced biliary secretion of BA and increased fecal excretion of BA via decreased apical sodium-dependent bile salt transporter (Asbt)-mediated reabsorption. Atorvastatin decreased liver steatosis and inflammation in NASH animals consistently with a reduction in crucial lipogenic enzyme stearoyl–coenzyme A (CoA) desaturase-1 and nuclear factor kappa light chain enhancer of activated B-cell pro-inflammatory signaling, respectively. In this group, atorvastatin also uniformly enhanced plasma concentration, biliary secretion and fecal excretion of the secondary BA, deoxycholic acid (DCA). However, in the chow diet–fed animals, atorvastatin decreased plasma concentrations of BA, and reduced BA biliary secretions. These changes stemmed primarily from the increased fecal excretion of BA resulting from the reduced Asbt-mediated BA reabsorption in the ileum and suppression of synthesis in the liver. In conclusion, our results reveal that atorvastatin significantly modulates BA metabolomics by altering their intestinal processing and liver synthesis in control and NASH mice.

## 1. Introduction

Nonalcoholic fatty liver disease (NAFLD) is the most common liver disorder worldwide, affecting approximately 30% to 40% of men and 15% to 20% of women. The major form is characterized by simple steatosis, but 30-40% of affected patients develop nonalcoholic steatohepatitis (NASH), which is characterized by steatosis, necroinflammatory changes, and varying degrees of liver fibrosis [1]. As an aggressive form of NAFLD, NASH raises the overall mortality rate by 57%, with deaths primarily linked to liver-related and cardiovascular (CVS) causes [2]. Currently, the best therapeutic intervention is the reduction of one’s weight by way of lifestyle changes in dietary and exercise habits, but this approach is commonly unsuccessful. Therefore, it is necessary to study all factors that may modulate the development of NASH, in order to enable effective therapeutic intervention. Bile acids (BA) are molecules with a significant function in the pathophysiology of NASH [3]. These endogenous amphipathic steroids act through specific receptors, such as the farnesoid X (FXR) (*NR1H4*), pregnane X (*NR1I2*), or G-protein-coupled (TGR5 and sphingosine-1-phosphate receptor 2) receptors; inhibit hepatic lipogenesis, gluconeogenesis, and inflammation; and promote energy expenditure. However, these effects require the intact function of BA transporters to conserve and compartmentalize BA at high concentrations within the intestinal and hepatobiliary tracts and to restrict BA systemic exposure. As such, the inflammation accompanying NASH impairs BA excretion into the bile, triggering increased systemic exposure and promoting cumulative BA toxicity [4]. It is, therefore, important to study the modulation of BA homeostasis by drugs that are considered eligible for the treatment of NASH itself or are used as therapeutics to address its metabolic and CVS complications.

The incidence of NASH is increased in patients with type 2 diabetes mellitus and with morbid obesity [5]. The typical accompanying complications include hypercholesterolemia with atherosclerosis and related CVS complications [6]. Such patients are therefore frequently treated by statins, i.e., 3-hydroxy-3-methyglutaryl coenzyme A reductase (HMGCR) inhibitors, which are the principal drugs used for reducing serum cholesterol concentrations. In addition to the beneficial effects of statins on lipids, these drugs also improve insulin sensitivity, reduce the production of advanced glycation end-products, and display anti-inflammatory effects, all of which may be helpful in treating steatosis and inflammation associated with NASH [7]. Indeed, data from large prospective randomized clinical trials suggest that atorvastatin especially ameliorates NAFLD/NASH and reduces CVS events by two fold relative to those with normal liver function [8]. Consistent with these findings, several animal studies have shown that statins attenuate NAFLD and decrease liver inflammation, steatosis, and fibrosis [9,10,11,12,13,14,15,16,17]. Molecular studies have suggested that the mechanisms of statins’ protective activity in NASH include the activation of the adenosine monophosphate–activated protein kinase α (AMPKα) and the inhibition of c-Jun N-terminal kinases (JNK1/2), Rho kinase [18], and liver X receptor α–related genes [19]. However, the available data remain inconclusive at this time.

The impact of statins on BA homeostasis has been studied in healthy animals. Under these circumstances, statins reduced the liver and intestinal contents of several BA [20,21] and increased fecal BA excretion [20,22,23]. Together with simultaneous changes in messenger RNA (mRNA) expression of enzymes and transporters mediating BA turnover, their results indicate that statins have the potential to alter BA homeostasis in liver pathologies such as NASH. However, comprehensive data concerning the effects of statins on BA turnover during NASH are missing. Similarly, no study has explored the influence of statins on the biliary secretions of individual BA in healthy animals. It is of interest to mention that, despite the significant posttranscriptional modulation of crucial transporters for enterohepatic recycling of BA during NASH, available studies have demonstrated the expression of relevant molecules mainly at the mRNA level [24,25,26]. Therefore, the present study characterized concentrations of individual BA in plasma, bile, and feces of mice with NASH induced by 24 weeks of consumption of a high–saturated fat, high-fructose, and high-cholesterol diet (F), with atorvastatin administered during the last three weeks. Significant differences were found in BA metabolism in the atorvastatin-treated group.

## 2. Results

### 2.1. Atorvastatin Mitigated Hepatic Steatosis and Inflammation in a NASH Mouse Model

The F diet, as a combination of a high-fat diet with glucose/fructose in drinking water, provoked liver steatosis and the development of inflammation and fibrosis in mice [27]. To induce NASH, C57BL/6 male mice were fed an F diet over 24 weeks. In comparison with a chow diet, the F diet induced significant elevations in body weight; liver weight; liver to body weight ratio (%); serum alanine aminotransferase (ALT), an indicator of hepatocellular injury; alkaline phosphatase (ALP), a marker of cholestatic liver impairment; triglycerides (TG); and soluble endoglin (Figure 1)—a recently suggested marker of liver fibrosis in NASH [28]. Atorvastatin effectively reduced plasma ALT and TG concentrations (Figure 1), indicating its positive effect on liver injury.

The mice fed with the F diet developed histopathological features of NASH with macrovesicular steatosis, ballooning, inflammatory cell infiltration, and fibrosis as demonstrated by hematoxylin and eosin and Sirius red staining (Figure 2). The increased liver infiltration by inflammatory cells in the NASH group was further confirmed by the significantly heightened protein levels of the pro-inflammatory transcription factor nuclear factor kappa light chain enhancer of activated B-cells (NF-κB) p65, and F4/80 markers of Kupffer cells plus recruited macrophages (Figure 2). Consistent with the reduced ALT and TG levels, atorvastatin-treated NASH animals showed significant attenuation of macrovesicular liver steatosis and inflammatory cell infiltration as detected by histological evaluation and by the significantly reduced NF-κB p65, and F4/80 protein expression levels (Figure 2). At this stage, however, atorvastatin could not reduce the total liver content of TG (Figure 2). Moreover, atorvastatin did not mitigate liver fibrosis as evaluated by Sirius red liver staining and by the mRNA expression of fibrotic markers such as *αSMA*, *Col1a1*, and *Tgf**β1* (Figure 2). Importantly, atorvastatin did not impair any biochemical or histological liver parameters in animals fed a chow diet. 

These findings indicate the successful induction of NASH by the F diet and its attenuation by atorvastatin via the reduction of macrovesicular steatosis and inflammation.

### 2.2. Atorvastatin Reduced Liver TG Synthesis in NASH Mice

We then analyzed the hepatic gene expression of molecules crucial for TG uptake, metabolism, storage, and synthesis to elaborate the mechanism of atorvastatin-mediated alleviation of hepatic steatosis (Figure 3). Consistent with the biochemical and histological features of liver injury, vehicle-treated NASH mice demonstrated significantly increased mRNA of *CD36*, an uptake transporter for TG, and stearoyl-CoA desaturase-1 (*Scd1*), a key enzyme in fatty acid (FA) synthesis, and a reduction in microsomal TG transfer protein (*Mttp*), the transporter necessary for lipid export and assembly of lipoproteins. Atorvastatin intervention in NASH animals significantly attenuated only the expression of *Scd1*. Atorvastatin did not modulate the expression of the majority of molecules in chow diet–fed animals but did increase the expression of uptake and lipogenic genes, *CD36*, FA synthase (*Fasn*), and acetyl-CoA carboxylase (*Acc*). These data indicate that atorvastatin administration to F animals reduced the gene expression of *Scd1*, the enzyme crucial in liver monounsaturated FA synthesis.

### 2.3. Atorvastatin Modified BA Spectra in Lean and NASH Mice

Statins may reduce BA disposition in healthy lean mice likely by way of the reduced synthesis of their precursor, cholesterol, and also by the tendency to enhance their loss via feces [20]. However, to elaborate these mechanisms, we measured plasma concentrations of BA in atorvastatin-administered chow and F diet–fed mice. Plasma concentrations of BA in vehicle-treated lean mice showed a dominant proportion of non12α-OH, primary, and conjugated BA (Figure 4). Notably, the most abundant BA were (T)MCA, (T)CA, and (T)DCA (Figure 4). Atorvastatin-treated lean mice displayed a decrease in total plasma BA as compared with vehicle-administered controls, with (T)MCA being responsible for this reduction. Consequently, CA mice showed an increased ratio of 12α-OH/non12α-OH BA and a reduced ratio of primary/secondary BA, respectively (Figure 4). As compared with the control group, vehicle-administered NASH mice showed unchanged total BA plasma concentrations, but secondary BA were increased with raised 12α-OH/non12α-OH and reduced primary/secondary ratios, respectively (Figure 4). This effect was caused by a proportional change in BA concentrations. Only the (T)CDCA concentration was significantly increased in F diet–treated mice, consistent with a previous report [30]. An increased plasma level of (T)CDCA has been shown to correlate with the severity of hepatic steatosis in a high-fat, high-cholesterol NAFLD model [31]. Atorvastatin-treated NASH mice (FA group) achieved a significant reduction in the primary/secondary BA ratio and conjugated/unconjugated BA ratio relative to untreated F mice (Figure 4). These changes are correlated especially with the increased concentrations of DCA and minor UDCA in FA mice in comparison with the F diet group.

### 2.4. Atorvastatin Reduced BA Biliary Secretions in Lean Mice

To explore whether atorvastatin modifies bile secretion, we analyzed BA and glutathione in bile collected after cannulation of the gallbladder (Figure 5). Atorvastatin significantly reduced bile flow and biliary secretion of BA in lean mice, while biliary secretion of glutathione remained unaffected relative to untreated controls (Figure 5). Atorvastatin significantly reduced biliary secretions of non12α-OH, primary, and conjugated BA, which corresponded with a reduction in (T)MCA, and minor BA, (T)UDCA, and (T)HCA. There were no differences in the biliary secretions of 12α-OH BA, which led to an increased 12α-OH/non12α-OH BA ratio. Bile flow and BA biliary secretion were significantly reduced in F mice. An analysis of BA ratios demonstrated the increased proportion of conjugated over unconjugated BA despite that the biliary secretions of both groups were reduced. Atorvastatin therapy in F mice had no significant effect on bile flow or biliary secretion of total BA when compared with the untreated F group. However, atorvastatin selectively increased biliary secretions of secondary TDCA and subsequently reduced the primary/secondary BA ratio.

### 2.5. Atorvastatin Increased BA Fecal Excretion in Lean Mice

BA were analyzed in feces collected within 24 h. Detected BA were mainly unconjugated DCA, βMCA, and CA (Figure 6). The administration of atorvastatin to control mice increased fecal BA excretion, mainly by increasing the excretion of (T)DCA, when compared with the untreated group (Figure 6). Fecal excretions of less abundant LCA, and HCA were also increased by atorvastatin. The increased presence of DCA in the feces of CA mice led to an increased 12α-OH/non12c-OH ratio. An analysis of fecal BA in vehicle-administered NASH revealed significantly increased fecal excretion of total BA relative to untreated lean mice. Excretion of all three major BA—namely, DCA, βMCA, and CA—was increased, with an increased proportion of unconjugated BA. The 12α-OH/non12α-OH and primary/secondary BA ratios remained unchanged between the C and F groups. Treatment with atorvastatin increased the fecal excretion of DCA and the 12α-OH/non12α-OH BA ratio in NASH animals relative to vehicle-administered littermates.

Furthermore, we analyzed essential BA transporters and regulators in the ileum to understand the mechanisms of the observed change in BA fecal elimination. The increase in BA fecal excretion in the atorvastatin-treated control group and both NASH groups paralleled repression of the apical sodium-dependent bile salt transporter (*Slc10a2*/Asbt), the major uptake transporter for the reabsorption of BA from the intestine lumen. In contrast, atorvastatin administration in NASH mice induced ileum mRNA expression of *Nr0b2/*Shp and *Fgf15*, which are major targets of the FXR nuclear receptor for BA, as compared with untreated vehicle NASH controls.

These findings together revealed distinct modulation of BA homeostasis by atorvastatin in healthy and NASH mice.

### 2.6. Protein Expression of BA Synthetic Enzymes and Transporters

To explain the changes in BA homeostasis induced by atorvastatin, we analyzed the expression levels of BA-processing proteins in the liver (Figure 7). The administration of atorvastatin to lean mice reduced the protein expression of Mrp4, a major basolateral efflux protein for BA, and reduced the protein expression of sterol-12-α-hydroxylase (Cyp8b1); sterol-27-hydroxylase (Cyp27a1); and Cyp2c70, a major rodent isoform responsible for MCA synthesis, relative to vehicle-fed littermates. Mice fed with the F diet showed a significant reduction in the protein expression levels of bile salt export pump (Bsep), the rate-limiting step in BA biliary secretion; sodium/taurocholate co-transporting polypeptide (Ntcp), a major uptake transporter for BA in the liver; Mrp4; cholesterol-7-α-hydroxylase (Cyp7a1), the rate-limiting enzyme in the classic pathway of BA synthesis; Cyp8b1; Cyp27a1; and Cyp2c70. Atorvastatin treatment increased the protein expression level of Mrp4 and reduced those of Cyp7b1 and Cyp2c70 in NASH mice.

### 2.7. Regulation Pathways for BA Homeostasis

Proteins responsible for BA turnover are regulated transcriptionally as well as posttranscriptionally. To further elaborate atorvastatin-mediated regulation, we analyzed the mRNA expression of BA-related enzymes and transporters (Figure 7). Interestingly, the mRNA expression of BA liver synthetic enzymes and transporters remained unchanged following atorvastatin treatment in lean mice, with the only exception being a moderate but significant induction of *Cyp8b1*. The F diet induced transcription of *Abcc2* (Mrp4) and repressed the transcription of *Cyp7a1*, *Cyp7b1*, *Cyp27a1*, *Cyp2c70*, *Slc10a1* (Ntcp), and *Abcc2* (Mrp2) mRNA as compared with vehicle-administered lean mice, which indicates that transcriptional regulation can be identified in F diet–fed animals for *Ntcp*, *Cyp7a1*, *Cyp27a1*, and *Cyp2c70* mRNA. Other BA-related enzymes and transporters showed a degree of discrepancy between mRNA and protein expression levels, suggesting posttranscriptional mechanisms. Atorvastatin treatment in the F group did not change the mRNA expression levels of either measured enzymes or transporters except for induction of *Abcb11* (Bsep). The discrepancies between the mRNA and protein expression levels of liver BA-related molecules in the atorvastatin-treated group indicated posttranscriptional regulation.

## 3. Discussion

The positive effects of statins on NASH have been reported to date in several animal and some human studies [8,32,33]. Our findings of improved liver architecture and plasma biochemistry in the atorvastatin-treated NASH group are consistent with these data. Previous experimental studies also demonstrated improved liver fibrosis after statin administration [34,35] which was not detected in our study. However, in a corresponding high-fat high-cholesterol diet-induced model of steatosis, 8 weeks of atorvastatin administration were necessary to reduce liver fibrosis. Therefore, we anticipate that 3 weeks of atorvastatin administration that identified primary steps of its hepatoprotective mechanisms in NASH were too short to develop antifibrotic effect. However, none of the existing articles presented a comprehensive evaluation of BA metabolomics in statin-treated NASH animals. He et al. [30] only analyzed changes in the relative abundance of serum BA, where simvastatin reduced the proportions of (T)CDCA, DCA, and TUDCA and increased the percentage of LCA in mice with simple liver steatosis. All mechanisms identified in our study, for atorvastatin in NASH animals, are summarized in Figure 8.

We detected that atorvastatin did not change the absolute concentration of total BA in the plasma of NASH mice. This corresponds with the sole observation that statins may not alter the plasma concentration of total BA in patients with NASH [4]. However, we found reduced plasma primary/secondary BA ratios and conjugated/unconjugated BA ratios in NASH mice. These were caused by the increased concentrations of (T)DCA and UDCA. Similarly, atorvastatin reduced the primary/secondary BA ratio in bile via increased (T)DCA. DCA was the only BA increased by atorvastatin in the stool of NASH animals. The uniform increase in the concentration of DCA, the FXR receptor agonist, in bile and stool is consistent with the enhanced expression of FXR target genes such as *Nr0b2/*Shp and *Fgf15* in the ilea of NASH mice. Further studies are needed to unravel whether DCA may be an indicator of statin action in NASH.

The increased fecal excretion and overall content of DCA in the atorvastatin-administered NASH group are in agreement with the previously described, increased relative DCA abundance in the cecum of simvastatin-treated rats with simple liver steatosis [36]. This suggests that statins increase the synthesis of secondary BA by modulating gut microbiota. Indeed, therapy with statins reduces the bacterial diversity of the gut microbiota in obese humans [37] and mice [21], increases the relative abundance of the genera *Bacteroides* [38] and *Proteobacteria,* and reduces the abundance of *Firmicutes* [30,39]. Although increased DCA production usually corresponds with the expansion of *Firmicutes* via 7α/β-dexydroxylation—especially by *Clostridium leptum* [40]—*Bacteroides* also promotes the conversion of primary BA into secondary BA through bile salt hydrolase activity [41]. The reduced ratio of conjugated to unconjugated BA in our CA mouse group supports this hypothesis. However, an increased DCA concentration is also consistent with the suppression of enzymes necessary for the acidic pathway of BA synthesis, especially Cyp7b1, and Cyp2c70, which may cumulate precursors for neutral BA synthesis such as DCA. Details of such regulation are currently unknown, but the data together suggest the occurrence of complex regulation of BA metabolomics by statins.

Animal and in vitro studies analyzing the mechanisms of NASH attenuation by statins collectively have ascribed this effect to the suppression of liver inflammation and fibrosis [32]. Statins reduce hepatic lipid macrophage recruitment and activation [17] via the reduction of mevalonate-derived molecules such as farsenyl pyrophosphate, which is necessary for prenylation of small GTPases such as RhoA/Rho-kinase, Ras, or RAC1, enabling their cell membrane anchoring and activation [18]. Other protective mechanisms of statins involve the inactivation of JNK [17], NgBR, ERK1/2, or Akt [19]. We confirmed this effect through reduced F4/80 macrophage markers and NF-κB pro-inflammatory transcription factors. However, less is known about the mechanisms involved in the reduction of liver steatosis by statins [10,11,34], which was not detected in some studies [17,42]. There are discrepancies in reported mechanisms, in that statins may either restore peroxisomal FA β-oxidation [11,43] or attenuate activated FA synthesis in steatotic livers via reduced FA synthase (*Fasn*) and acetyl-CoA carboxylase (*Acc*) [10]. The latter mechanism is consistent with the inhibition of hepatic steatosis in *ob/ob* mice with deleted Rho-kinases through the repression of *Fasn* and *Scd1* (stearoyl desaturase-1) without an influence on FA β-oxidation. This concept is supported by the finding of reduced *Scd1* mRNA in our FA group. The isolated reduction in *Scd1* mRNA within three weeks of therapy by atorvastatin may indicate the initial point of its effect on FA hepatic homeostasis in NASH mice. The primary effect of statins on Scd1 is also supported by a previously reported isolated reduction in liver monounsaturated free fatty acids in high-fat diet–fed wild-type mice treated with atorvastatin [17].

In addition to the positive liver effects of atorvastatin in mice with NASH, we also demonstrated that atorvastatin decreased plasma concentrations of BA in lean mice despite the reduction in their biliary secretions. This suggests reduced BA availability, which is consistent with the decreased BA pool size described in a previous study, albeit without a change in plasma BA concentrations [20]. Fu et al. [20] anticipated that the mechanism of BA pool size reduction by statins may involve the reduction of cholesterol disposition for BA synthesis. The same study also demonstrated a tendency to increase BA fecal elimination yet failed to reach statistical significance, perhaps due to the smaller experimental groups and short duration of therapy used. The mRNA expression of ileal BA transporters was also unchanged [20]. In the present study, we did not detect reduced cholesterol in the liver (data are not shown), similarly to previous studies with a corresponding dose of atorvastatin [17,42,44]. This result may be explained by the fact that statins primarily control levels of LDL cholesterol, which, compared to humans, is a minor component in plasma lipoproteins of mice [23]. In contrast, we indeed demonstrated a statistically significant increase in the fecal excretion of BA using an adequate study population and three weeks of drug administration. BA fecal excretion increased despite the reduced BA delivery to the intestine via bile, which strongly suggests a reduced amount of BA reabsorption occurring from the intestine in the CA group. Consequent analysis of Asbt, the major transporter for BA reabsorption from the ileum, revealed the repressing effect of atorvastatin on its expression, for the first time. This is a typical consequence of activated FXR signaling, which may be caused by the increased concentrations of DCA and reduced delivery of antagonistic MCA. Taken together, our findings suggest the ability of statins to reduce plasma BA concentrations by reducing intestinal reabsorption in individuals with intact liver function.

Atorvastatin reduced the bile flow in the lean mice (CA group). The biliary secretion of glutathione, a major component of BA-independent bile flow, was unchanged in these mice, consistent with no change in the liver Mrp2 protein concentration. This signifies that the reduction of bile flow was BA-dependent. Indeed, biliary secretion of BA was reduced in CA animals but without a change in the protein expression levels of Bsep and Ntcp. A similar absence of the change in Bsep expression was observed in previous studies of mRNA [20,44], but also protein level [45]. Taken together, the reduced BA-dependent bile flow in our CA animals may be the consequence of a reduced disposition of BA due to their reduced reabsorption in the ileum. Furthermore, the reduction of BA biliary secretions in atorvastatin-treated lean mice was not proportional for all BA and included TMCA and βMCA. This suggests that the reduced liver synthesis of these primary BA contributes to the reduced BA biliary secretion and subsequently to the reduced bile flow. This is consistent with the previously reported reduction of MCA in the livers of atorvastatin-treated mice [20]. Therefore, we analyzed the expression level of Cyp2c70, the enzyme responsible for muricholic acid synthesis in rodents, and its downregulation by atorvastatin was detected. The mechanism of Cyp2c70 regulation is currently unknown. Similarly, the interpolation of this finding to humans is a major limitation of this study because, unlike rodents, humans do not synthesize MCA in significant quantities. However, the reduction of Cyp8b1 and Cyp27a1 may have relevancy for humans.

The majority of available studies indicate that statins increase *Cyp7a1* mRNA expression in rodents [19,20,44,46], likely due to the decreased FXR signaling [20]. It was anticipated that increased expression of *Cyp7a1* mRNA compensates for the reduced synthesis of FXR agonistic BA such as CA and DCA due to the statin-induced blockade of cholesterol synthesis [20]. However, our study together with some other research demonstrated that statins may not have an effect on *Cyp7a1* mRNA expression [21] or that this enzyme may even be reduced [22]. The discrepancy is supported by discordant changes in the mRNA levels of other FXR target genes such as *Abcb11* (Bsep), *Slc10a1* (Ntcp), *Slc10a2* (Asbt), and *Slc51a/b* (Ostα/β) [20,44,45]. The cause of such diversity may be the variability of dosage schedules tested in different species. Indeed, an analysis of dose-dependency uncovered a culminating effect of atorvastatin on the mRNA levels of Bsep, Ntcp, Cyp7a1, and Shp at the dose of 30 mg/kg, with its attenuation at 100 mg/kg [46]. Thus, our study presents the mid-term effects of statins given at a submaximal effective dosage.

In summary, this study mainly elucidated mechanisms of altered BA enterohepatic recycling in atorvastatin-treated mice with intact and NASH livers. Our results highlighted the increased BA loss that occurs through the feces, leading to reduced BA biliary secretions, and the decreased systemic plasma concentrations found in mice with intact livers. These changes are linked to the downregulation of Asbt in the ileum and reduced synthesis in the liver. Atorvastatin demonstrated a therapeutic effect on NASH, with significant improvements in serum ALT, liver steatosis, and inflammation. For the first time, we have demonstrated changes in BA metabolomics involving especially a greater turnover of the secondary BA, DCA. The potential role of DCA as an indicator of statins’ liver effect in NASH should be further clarified.

## 4. Materials and Methods

### 4.1. Animals and Diets

All animals received care according to the guidelines set by the Animal-Welfare Body of Charles University (Prague, Czech Republic). Six-week-old male C57BL/6 mice were purchased from Velaz (Prague, Czech Republic). All animals were housed in individually ventilated cages at a constant humidity level of 55 ± 10% and a temperature of 23 ± 1 °C under a 12-h light/dark cycle. Animals had free access to water and food. After a one-week adaptation period, the mice were randomly assigned into two experimental groups (*n* = 16 mice each): the mice were fed a chow diet (C, PicoLab^®^ RD 20, LabDiet) and tap water or the F diet consisting of a Western diet (AIN-76A WD, 20% of fat; TestDiet, St. Louis, MO, USA) and glucose (18.1 g/L) with fructose (24 g/L) provided in water over 21 weeks. The F diet was 40% kcal from fat (milk fat), 0.2% from cholesterol, 15.4% from protein (casein), and 44.5% from carbohydrates (sucrose). The animals were then randomly allocated into the following four groups (*n* = 8 mice each): a control lean diet group on a chow diet (C), a control plus 20 mg/kg of atorvastatin (CA) group, an F diet group (F), and an F diet plus 20 mg/kg of atorvastatin (FA) group. Atorvastatin or vehicle (1% methylcellulose) was administered once daily by oral gavage for another three weeks together with the respective diets. Immediately following the administration of the last dose of atorvastatin or vehicle, the mice were placed in metabolic cages, and stool was collected for 24 h. The mice were then anesthetized by intraperitoneal pentobarbital (50 mg/kg), and bile was collected for 45 min through the cannulated gallbladder. At the end of the experiment, a blood sample was taken, the mice were sacrificed by anesthesia, and their livers and ileum excised and immediately frozen in liquid nitrogen. Plasma samples were obtained from the whole blood by centrifugation at 2000× *g* for five minutes at 4 °C. Samples were stored at −80 °C until analysis. Selection of atorvastatin for our study was based on a series of results, that suggested its efficacy in alleviating NASH in preclinical and clinical studies [32]. The dose [17,47] and timing [42] of atorvastatin were selected on the basis of previous research and following analyses of drug tolerance in preliminary studies.

### 4.2. Analytical Methods

Liver enzymes were detected in plasma using a commercial Preventive Care Profile Plus test with a Vetscan 2 device (Abaxis, Inc., Union City, CA, USA). Plasma and liver concentrations of triglycerides were analyzed with a commercial kit (Erba Lachema s.r.o., Brno, Czech Republic). BA concentrations in plasma, bile, and feces were measured by liquid chromatography–mass spectrometry as described previously [48]. The concentrations of individual BA were summed to derive the concentration of conjugated, unconjugated, and total BA, respectively. Primary BA included (tauro/glyco)cholic acid [(T/G)CA], (tauro)chenodeoxycholic acid [(T)CDCA], (tauro)-α-muricholic acid [(T)αMCA], and (tauro)-β-muricholic acid [(T)βMCA], while secondary BA include (tauro)deoxycholic acid [(T)DCA], (tauro)lithocholic acid [(T)LCA], (tauro)ursodeoxycholic acid [(T)UDCA], (tauro)murideoxycholic acid [(T)MDCA], (tauro)muricholic acid [(T)MCA], taurohyocholic acid (THCA), and (tauro)hyodeoxycholic acid [(T)HDCA]. The 12α-OH BA consist of (T)CA and (T)DCA, while non–12α-OH BA include all the remaining BA. Plasma concentrations of mouse soluble endoglin (sEng) levels were analyzed using the mouse endoglin/CD105 quantikine enzyme-linked immunosorbent assay kit (MNDG00; R&D Systems, Minneapolis, MN, USA) according to the manufacturers’ instructions, in duplicate.

### 4.3. Quantification of Gene and Protein Expression Levels

Reverse-transcription polymerase chain reaction (qRT-PCR) was conducted via the Quantstudio 7 HT fast RT-PCR system (Applied Biosystems, Foster City, CA, USA) as described previously [28]. The primers used for analysis are specified in Table 1. Western blotting was performed as reported previously [28]. The list of antibodies is presented as Table 2.

### 4.4. Histopathological Examination of the Liver

Sagittal sections of medial liver lobes were fixed in a 4% buffered formaldehyde solution embedded in paraffin and stained with hematoxylin and eosin for liver injury grading and with Sirius red to classify the degree of fibrosis, respectively. The severity of hepatic histological changes was assessed and scored in a blinded manner using the NASH Clinical Research Network scoring system [29]. The steatosis was scored according to the percentage of parenchymal involvement. The lobular inflammation grade was scored by the numbers of inflammation foci in the area of ×200 microscopic fields as follows: grade 0, no foci; grade 1, less than two foci per ×200 field; grade 2, two to four foci per ×200 field; and grade 3, more than four foci per ×200 field. The ballooning was scored by the number of ballooned hepatocytes per visual field: 0, none; 1, few ballooned cells; 2, many cells affected or prominent injury with patches of cells. The fibrosis stage was scored according to the location and density of the fibrosis as follows: stage 0, none; stage 1, perisinusoidal or periportal fibrosis; stage 2, perisinusoidal and periportal fibrosis; stage 3, bridging fibrosis; and stage 4, cirrhosis.

### 4.5. Statistical Analysis

All statistical analyses were conducted using the GraphPad Prism 8 statistical software (GraphPad Software, San Diego, CA, USA). Data are presented as median values, with boxes and whiskers representing the interquartile range and fifth to 95th percentiles, respectively, according to the data distribution. The statistical significance (*p* < 0.05) was determined using either one-way analysis of variance for continuous variables or one-way analysis of variance on ranks for nonparametric variables.

## Figures and Tables

**Figure 1 ijms-22-06468-f001:**
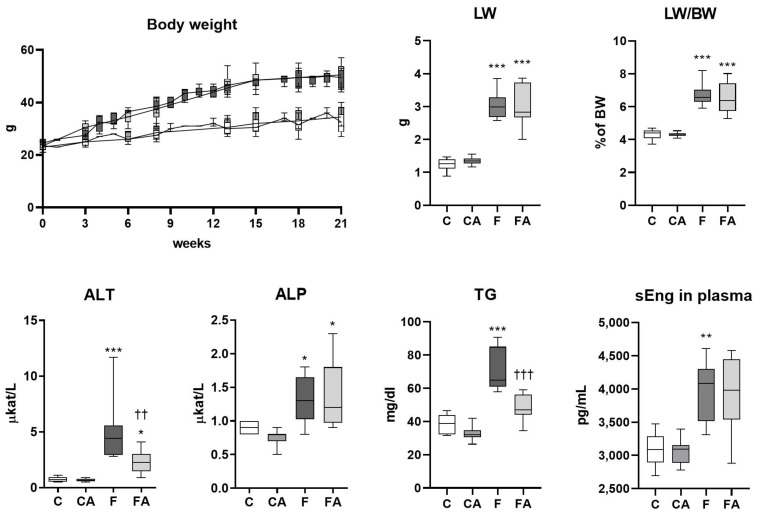
Atorvastatin treatment ameliorated the increased levels of ALT and TG found in the plasma of mice fed a NASH-inducing diet. NASH was induced in mice by 24 weeks of F diet consumption, with atorvastatin (A) administered orally (20 mg/kg/day) during the last three weeks. Corresponding control groups received a chow (C) diet. Abbreviations: BW—body weight, LW—liver weight, ALT—alanine aminotransferase, ALP—alkaline phosphatase, TG – triglycerides, sEng—soluble endoglin. Data are presented as median values, with boxes and whiskers representing the interquartile range and fifth to 95th percentiles, respectively. * *p* < 0.01, ** *p* < 0.01, and *** *p* < 0.001 atorvastatin and/or F-diet vs. vehicle-administered chow diet–treated mice; †† *p* < 0.01 and ††† *p* < 0.001 atorvastatin-treated vs. vehicle-treated NASH mice.

**Figure 2 ijms-22-06468-f002:**
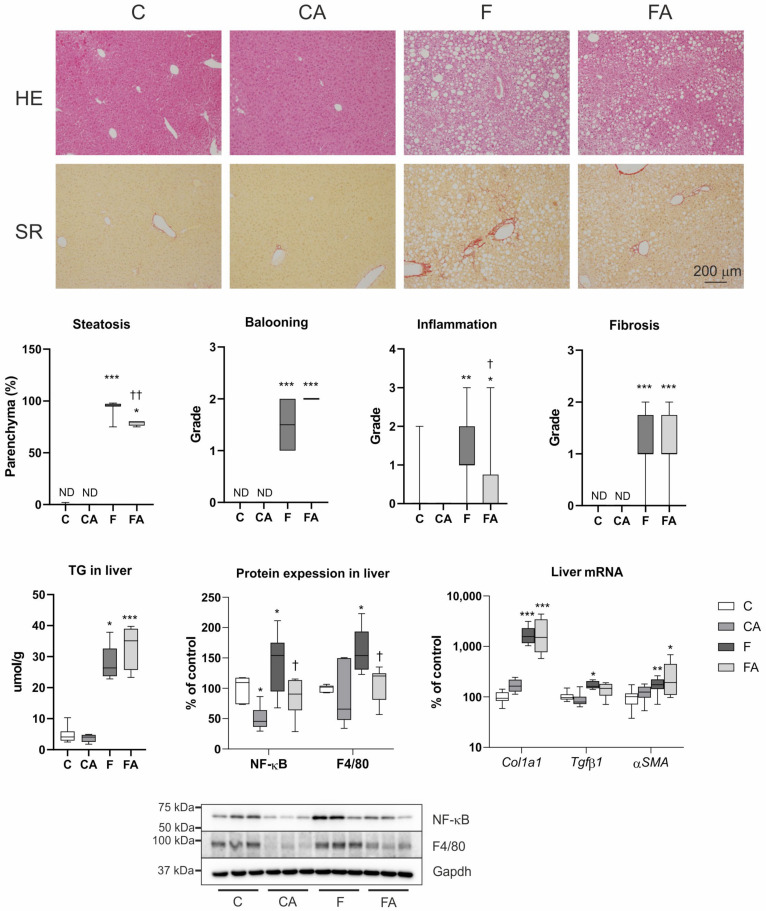
Atorvastatin treatment reduced steatosis and inflammation in mice with NASH. Representative hematoxylin and eosin (HE) and Sirius red (SR) staining (magnification **×**100). NASH was induced in mice by 24 weeks of F diet consumption, with atorvastatin (A) administered orally (20 mg/kg/day) during the last three weeks. Corresponding control groups received a chow (C) diet. The steatosis was scored according to the percentage of parenchymal involvement. The lobular inflammation, ballooning, and the fibrosis were graded according to the NASH Clinical Research Network scoring system [29]. Abbreviations: ND—not detected. Data are presented as median values, with boxes and whiskers representing the interquartile range and fifth to 95th percentiles, respectively. * *p* < 0.01, ** *p* < 0.01, and *** *p* < 0.001 atorvastatin and/or F-diet vs. vehicle/chow diet–treated mice; † *p* < 0.05 and †† *p* < 0.01 atorvastatin-treated vs. vehicle-treated NASH mice.

**Figure 3 ijms-22-06468-f003:**
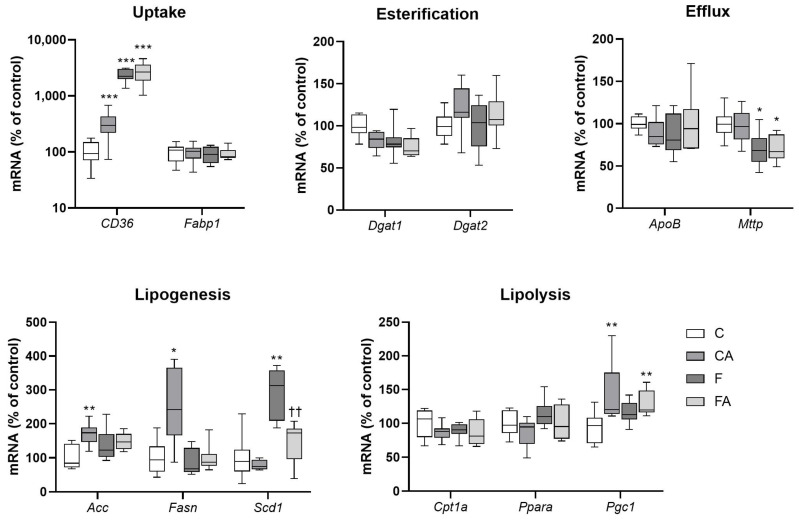
Atorvastatin and NASH modulated the expression levels of lipid metabolism genes in the liver. NASH was induced in mice by 24 weeks of F diet consumption, with atorvastatin (A) administered orally (20 mg/kg/day) during the last three weeks. Corresponding control groups received a chow (C) diet. Data are presented as median values, with boxes and whiskers representing the interquartile range and fifth to 95th percentiles, respectively. * *p* < 0.01, ** *p* < 0.01, and *** *p* < 0.001 atorvastatin and/or F-diet vs. vehicle-administered chow diet–treated mice; †† *p* < 0.01 atorvastatin-treated vs. vehicle-treated NASH mice.

**Figure 4 ijms-22-06468-f004:**
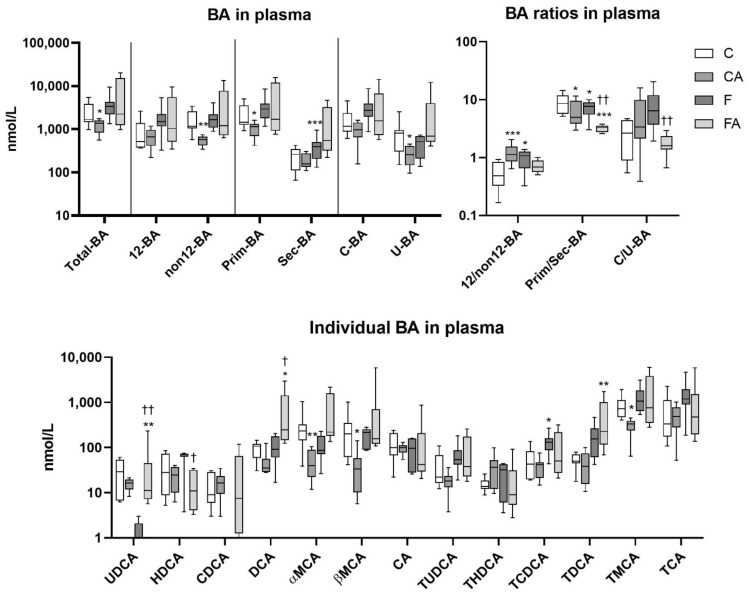
F diet consumption and atorvastatin modulated BA concentrations in plasma. NASH was induced in mice by 24 weeks of F diet consumption, with atorvastatin (A) administered orally (20 mg/kg/day) during the last three weeks. Corresponding control groups received a chow (C) diet. Data are presented as median values, with boxes and whiskers representing the interquartile range and fifth to 95th percentiles, respectively. * *p* < 0.01, ** *p* < 0.01, and *** *p* < 0.001 atorvastatin and/or F-diet vs. vehicle-administered chow diet-treated mice; † *p* < 0.05 and †† *p* < 0.01 atorvastatin-treated vs. vehicle-treated NASH mice. BA are presented as follows: 12α-hydroxylated (12 BA), non12α-hydroxylated (non12 BA), primary (Prim-BA), secondary (Sec-BA), conjugated (C-BA), unconjugated (U-BA), ursodeoxycholic acid (UDCA), hyodeoxycholic acid (HDCA), chenodeoxycholic acid (CDCA), deoxycholic acid (DCA), α/β muricholic acid (α/β MCA), and cholic acid (CA), with taurine conjugates TUDCA, THDCA, TDCA, TMCA, and TCA.

**Figure 5 ijms-22-06468-f005:**
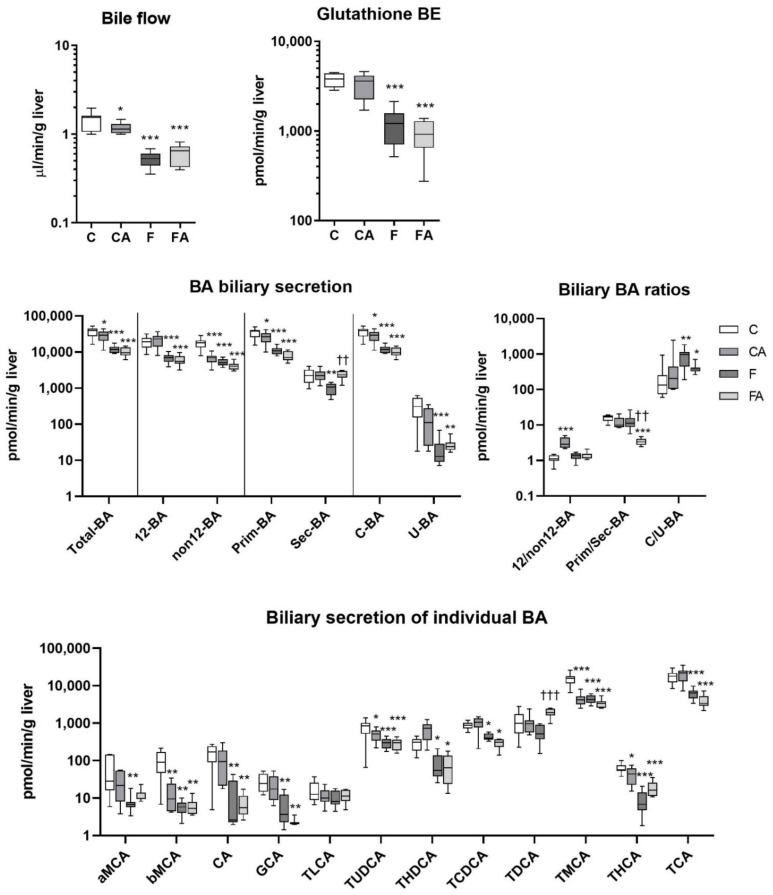
F diet consumption and atorvastatin administration modulated biliary excretion of BA. NASH was induced in mice by 24 weeks of F diet consumption, with atorvastatin (A) administered orally (20 mg/kg/day) during the last three weeks. Corresponding control groups received a chow (C) diet. BA and glutathione were analyzed in bile collected following cannulation of the gallbladder. Data are presented as median values, with boxes and whiskers representing the interquartile range and fifth to 95th percentiles, respectively. * *p* < 0.01, ** *p* < 0.01, and *** *p* < 0.001 atorvastatin and/or F-diet vs. vehicle-administered chow diet–treated mice; †† *p* < 0.01 and ††† *p* < 0.001 atorvastatin-treated vs. vehicle-treated NASH mice. BA are presented as follows: 12α-hydroxylated (12 BA), non12α-hydroxylated (non12 BA), primary (Prim-BA), secondary (Sec-BA), conjugated (C-BA), unconjugated (U-BA), α/β muricholic acid (α/β MCA), cholic acid (CA), glycocholic acid (GCA), taurolithocholic acid (TLCA), tauroursodeoxycholic acid (TUDCA), taurohyodeoxycholic acid (THDCA), taurochenodeoxycholic acid (TCDCA), taurodeoxycholic acid (TDCA), tauromuricholic acid (TMCA), taurohyocholic acid (THCA), and taurocholic acid (TCA).

**Figure 6 ijms-22-06468-f006:**
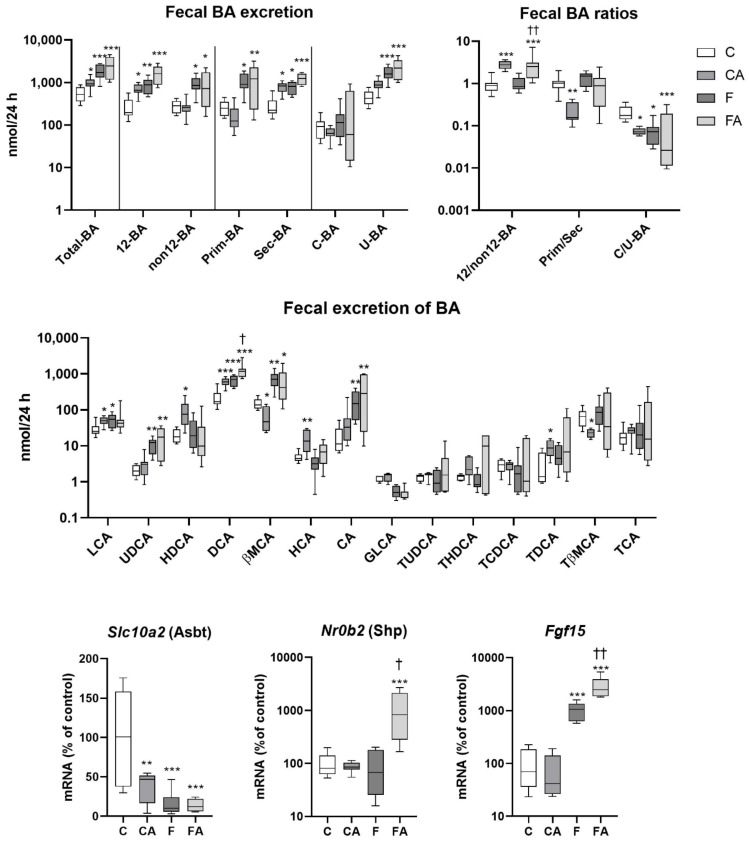
F diet consumption and atorvastatin treatment modulated the fecal excretion of BA. NASH was induced in mice by 24 weeks of F diet consumption, with atorvastatin (A) administered orally (20 mg/kg/day) over the last three weeks. Corresponding control groups received a chow (C) diet. BA were analyzed in the feces collected within 24 h. The mRNA expression levels of essential BA transporters and regulators were analyzed in ilea. Data are presented as median values, with boxes and whiskers representing the interquartile range and fifth to 95th percentiles, respectively. * *p* < 0.01, ** *p* < 0.01, and *** *p* < 0.001 atorvastatin and/or F-diet vs. vehicle-administered chow diet–treated mice; † *p* < 0.05 and †† *p* < 0.01 atorvastatin-treated vs. vehicle-treated NASH mice. BA are presented as follows: 12α-hydroxylated (12 BA), non12α-hydroxylated (non12 BA), primary (Prim-BA), secondary (Sec-BA), conjugated (C-BA), unconjugated (U-BA), lithocholic acid (LCA), ursodeoxycholic acid (UDCA), hyodeoxycholic acid (HDCA), deoxycholic acid (DCA), β muricholic acid (βMCA), hyocholic acid (HCA), cholic acid (CA), glycolithocholic acid (GLCA), tauroursodeoxycholic acid (TUDCA), taurohyodeoxycholic acid (THDCA), taurochenodeoxycholic acid (TCDCA), taurodeoxycholic acid (TDCA), tauro-β-muricholic acid (TβMCA), and taurocholic acid (TCA).

**Figure 7 ijms-22-06468-f007:**
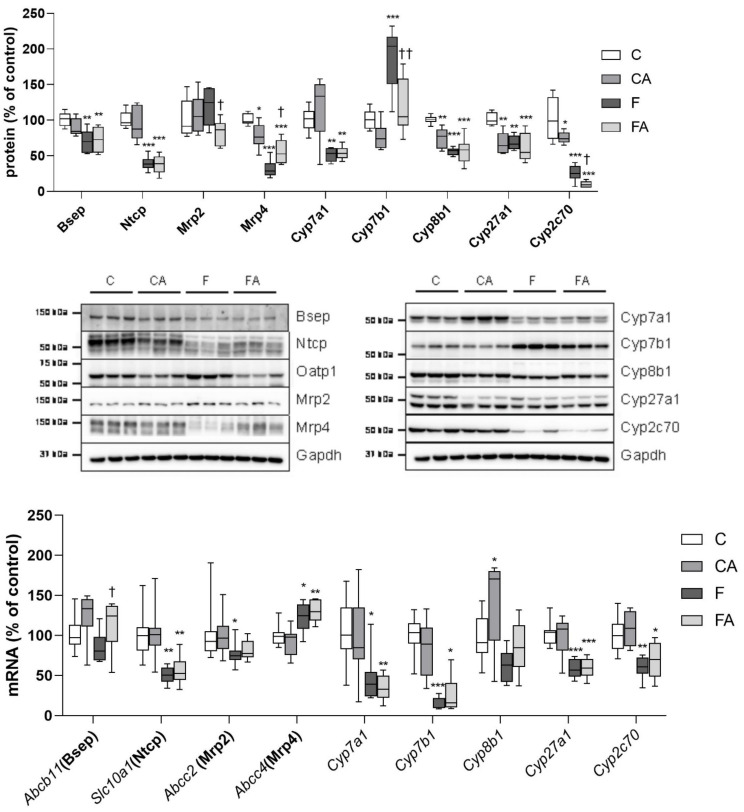
Influence of atorvastatin treatment and F diet consumption on the mRNA and protein expression levels of BA-related genes in the liver. NASH was induced in mice by 24 weeks of an F diet, with atorvastatin (A) administered orally (20 mg/kg/day) during the last three weeks. Corresponding control groups received a chow (C) diet. Data are presented as median values, with boxes and whiskers representing the interquartile range and fifth to 95th percentiles, respectively. * *p* < 0.01, ** *p* < 0.01, and *** *p* < 0.001 atorvastatin and/or F-diet vs. vehicle-administered chow diet–treated mice; † *p* < 0.05 and †† *p* < 0.01 atorvastatin-treated vs. vehicle-treated NASH mice.

**Figure 8 ijms-22-06468-f008:**
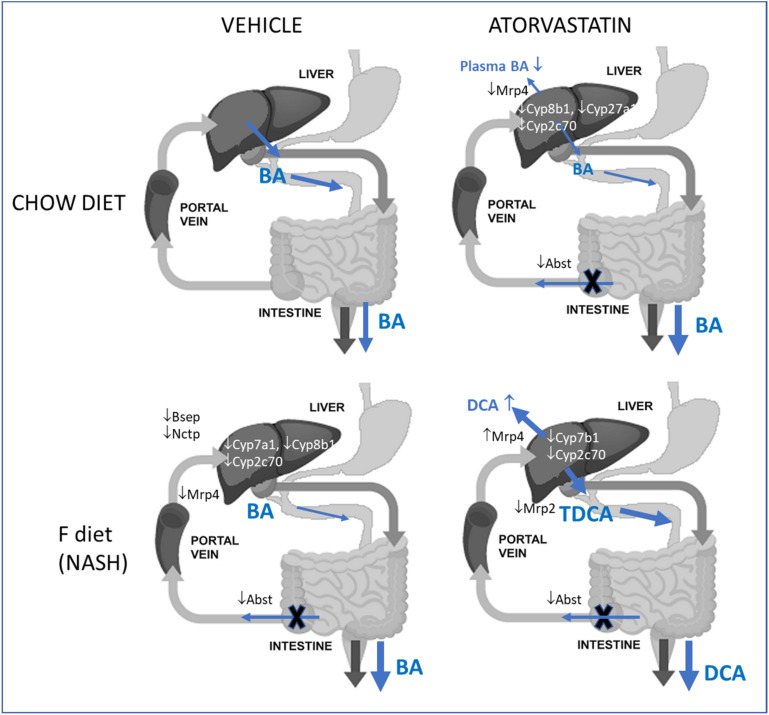
Possible mechanisms responsible for the effects of atorvastatin on BA metabolomics in mice with dietary-induced NASH. Changes induced by NASH and atorvastatin are depicted with arrows. The thickness of the blue arrow refers to the nature of change (increase or decrease) in the function vs. the corresponding untreated group. Abbreviations: BA—bile acids, Bsep—bile salt export pump; CD36—platelet glycoprotein 4, fatty acid translocase; DCA—deoxycholic acid; Mrp2/4—multidrug resistance–associated protein 2/4; Ntcp—Na^+^-taurocholate co-transporting polypeptide.

**Table 1 ijms-22-06468-t001:** Predesigned TaqMan^®^ gene expression assay kits (Life Technologies, Carlsbad, CA, USA) used for quantitative real-time RT-PCR.

Gene Symbol	Transporter/Receptor	Life Technologies Assay ID:
*Abcb11*	Bsep	Mm00445168_m1
*Slc10a1*	Ntcp	Mm00441421_m1
*Abcc2*	Mrp2	Mm00496899_m1
*Abcc4*	Mrp4	Mm01226380_m1
*Cyp7a1*		Mm00484150_m1
*Cyp7b1*		Mm00484157_m1
*Cyp27a1*		Mm00470430_m1
*Cyp8b1*		Mm00501637_s1
*Cyp2c70*		Mm00521058_m1
*Col1a1*		Mm00801666_g1
*Tgfb1*		Mm01178820_m1
*Acta2*	aSMA	Mm01546133_m1
*CD36*		Mm00432403_m1
*Acaca*	Acc	Mm01304257_m1
*Fasn*		Mm00662319_m1
*Scd1*		Mm00772290_m1
*Apob*		Mm01545150_m1
*Mttp*		Mm00435015_m1
*Slc10a2*	Asbt	Mm00488258_m1
*Nr0b2*	Shp	Mm00442278_m1
*Fgf15*		Mm00433278_m1
*Gapdh*		Mm99999915_g1

**Table 2 ijms-22-06468-t002:** Primary and secondary antibodies used in Western blotting.

Protein	Source	Dilution	Secondary Antibody Dilution
Cyp7a1	Sigma Aldrich (MABD42)	1:2000	1:5000
Cyp7b1	Bioss (Bs-5052R)	1:1000	1:3000
Cyp8b1	Thermo Fisher Scientific (PA5–37088)	1:1000	1:3000
Cyp27a1	Thermo Fisher Scientific (PA5–27946)	1:2000	1:5000
Cyp2c70	Mybiosource (MBS3223844)	1:1000	1:3000
Bsep	Thermo Fisher Scientific (PA5–13105)	1:1000	1:3000
Ntcp	Thermo Fisher Scientific (PA5-80001)	1:1000	1:3000
Mrp2	Thermo Fisher Scientific (PA5-49997)	1:2000	1:5000
Mrp4	Cell Signaling (12857S)	1:1000	1:3000
Nf-κB	Abcam (ab 16502)	1:1000	1:3000
Gapdh	Cell Signaling (2118L)	1:8000	1:10,000

## Data Availability

All data are contained within the manuscript. Only the results of liver cholesterol concentrations are mentioned in discussion as unpublished because there are no significant changes. These data are shared upon request by the corresponding author.

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
