# Peer review of "Atorvastatin Modulates Bile Acid Homeostasis in Mice with Diet-Induced Nonalcoholic Steatohepatitis"

_ijms, 2021, doi:10.3390/ijms22126468_

Round 1

Reviewer 1 Report

The study described in your manuscript has been properly designed and executed with data contributing to better understanding the potential therapeutic effects of atorvastatin on NASH. I recommend: a) improving the English style and punctuation in the abstract, b) Describe how you estimated the amount of ballooning ( I found only the score for steatosis, inflammation and fibrosis) and add the NASH scoring units along the y axis in fig 2. Mention in fig. 2 legends the method used to quantify histological parameters

Author Response

Dear reviewer,

Thank you very much for your quick review of our manuscript. All comments are important and below you find point-by-point answers, and suggested modifications performed in the manuscript.

Sincerely,

Stanislav Micuda

Reviewer 1:

The study described in your manuscript has been properly designed and executed with data contributing to a better understanding of the potential therapeutic effects of atorvastatin on NASH. I recommend:

  1. a) improving the English style and punctuation in the abstract,

The English was originally checked by a proofreading company (certificate enclosed), but we revisited the English again and tried to improve it.

  1. b) Describe how you estimated the amount of ballooning (I found only the score for steatosis, inflammation, and fibrosis) and add the NASH scoring units along the y axis in fig 2. Mention in fig. 2 legends the method used to quantify histological parameters

I am sorry for this, we have added the sentences in Methods: The steatosis was scored according to the percentage of parenchymal involvement. … The ballooning was scored by the number of ballooned hepatocytes per visual field: 0, none; 1, few ballooned cells; 2, many cells affected or prominent injury with patches of cells.

Figure 2 was elaborated to be more intelligible, and the description of methods for histological pictures quantification were added. The steatosis was scored according to the percentage of parenchymal involvement. The lobular inflammation, ballooning and the fibrosis was graded according the NASH Clinical Research Network scoring system (Kleiner et al., 2005).

Reference:

leiner DE, Brunt EM, Van Natta M, Behling C, Contos MJ, Cummings OW, Ferrell LD, Liu YC, Torbenson MS, Unalp-Arida A, Yeh M, McCullough AJ and Sanyal AJ (2005) Design and validation of a histological scoring system for nonalcoholic fatty liver disease. Hepatology 41:1313-1321.

Reviewer 2 Report

This is a review manuscript entitled “Atorvastatin modulates bile acid homeostasis in mice with diet-2 induced nonalcoholic steatohepatitis ” ” by Hana Lastuvkovaa, et al. In this study, the authors investigate a comprehensive evaluation of BA metabolomics in statin-treated NASH animals. They showed that atorvastatin reduce liver inflammation and steatosis in NASH model mice. Additionally atorvastatin demonstrated the changes in BA metabolomics involving especially a greater turnover of the secondary BA, DCA. This study is interesting and well-organized. This reviewer have some concerns below 1.Please show why the authors chose the atorvastatin, not other stains, including simvastatin. 2. Please discuss the reason why liver fibrosis did not be affected by atorvastatin administration, although inflamation and steatosis were restored in NASH mice, 3. Please discuss why the cholesterol was not changed by Atorvastatin (line 463-464). The dosage of admiration Atorvastatin is proper? Please described it 4. If possible, please show the changes in microbiota by atorvastatin administration. 5 In figure 1, the characters of “C” “CA” “F”, “FA” was overlap by the image of liver histology

Author Response

Dear reviewer,

Thank you very much for your quick review of our manuscript. All comments are important and below you find point-by-point answers, and suggested modifications performed in the manuscript.

Sincerely,

Stanislav Micuda

Reviewer 2:

This is a review manuscript entitled “Atorvastatin modulates bile acid homeostasis in mice with diet-2 induced nonalcoholic steatohepatitis ” ” by Hana Lastuvkova, et al. In this study, the authors investigate a comprehensive evaluation of BA metabolomics in statin-treated NASH animals. They showed that atorvastatin reduces liver inflammation and steatosis in NASH model mice. Additionally atorvastatin demonstrated the changes in BA metabolomics involving especially a greater turnover of the secondary BA, DCA. This study is interesting and well-organized. This reviewer have some concerns below

1.Please show why the authors chose the atorvastatin, not other stains, including simvastatin.

Our decision to use atorvastatin has several reasons. First, atorvastatin is the most frequently used statin in patients. Its prescription highly dominates over simvastatin due to better profile of therapeutic and adverse effects. Therefore, we considered results with atorvastatin as clinically more relevant. Second, a large part of clinical and preclinical studies relating statins and NASH were performed with atorvastatin (Athyros et al., 2017). This selection is not accidental and as it was recently discussed by Doumas et al (Doumas et al., 2018), current results suggest that the benefit of statins in NASH patients is compound specific and not a class effect. This is reinforced by studies in which simvastatin did not significantly affect NAFLD/NASH in patients (Nelson et al., 2009; Sarkar et al., 2016) while atorvastatin provided attenuation of these liver disorders (Athyros et al., 2006; Hyogo et al., 2012). The within-class differences in the effect of statins on NAFLD/NASH, could theoretically be attributed to the differences in the potency between members of this class of drug. Stronger statins, such as atorvastatin and rosuvastatin, might have greater ameliorating effects on NAFLD/NASH, whereas less potent statins might not show such benefits (Doumas et al., 2018).  In contrast, Park el al. (Park et al., 2016) have recently compared various statins, fluvastatin, pravastatin, simvastatin, atorvastatin, and rosuvastatin (15 mg/kg/day) in mice with NASH induced by a methionine- and choline-deficient diet. All these statins prevented the development of NASH with a similar potency. It might be therefore possible that rodents respond to statins better than humans. Taken together, selection of atorvastatin for our study was based on a series of results that suggested its efficacy in alleviating NASH in preclinical and clinical studies. Thus, our results will have higher relevance for interpretation of previous data. Third, atorvastatin was tested in previous studies using different dosage schedules. More results were therefore available for selection of proper dose. Finally, increased fecal excretion of bile salts was detected in mice treated with atorvastatin but no change was seen with other statins (Schonewille et al., 2016). 

Solution:

Following sentence was added to the methods section: Selection of atorvastatin for our study was based on a series of results that suggested its efficacy in alleviating NASH in preclinical and clinical studies (Athyros et al., 2017).

  1. Please discuss the reason why liver fibrosis did not be affected by atorvastatin administration, although inflammation and steatosis were restored in NASH mice.

Interestingly, such an effect was detected in a publication by Dongiovanni et al (Dongiovanni et al., 2015) where statin use was assessed for potential relationship with liver damage in patients with biopsy proven NASH. Statin use was associated with a reduced risk for hepatic steatosis and NASH; however, there was no difference regarding fibrosis. Similarly, The NAFLD Activity score, which incorporates steatosis, inflammation, and ballooning of hepatocytes was significantly reduced in another study , whereas 4/17 patients presented an increase in fibrotic stage (Hyogo et al., 2012). It seems that the dose of a statin plays a significant role in the degree of amelioration of NASH and specific histological lesions (Dongiovanni et al., 2015). As summarized by Athyros et al (Athyros et al., 2017) these effects were more pronounced in patients on statins not carrying the variant I148M of patatin-like phospholipase domain containing 3 (PNPLA3) genes. It is highlighted that the observed effect of the I148M variant of PNPLA3 gene on the severity of NAFLD/NASH is perhaps one of the strongest ever reported for a common variant modifying the genetic susceptibility of complex diseases. Furthermore, this variant is not associated with T2DM or IR. These observations may suggest that genetic factors might be involved in the degree of benefit from statin treatment on NASH amelioration.

Similar discrepancies exist in animal studies. Improved fibrosis have been reported in a just few studies mainly using methionine- and choline-deficient diets (Chong et al., 2015; Park et al., 2016), where even 4 week administration of fluvastatin reduced liver fibrosis. Much longer administration for 12 weeks was necessary to reduce fibrosis in rat model of high-fat and high-cholesterol diet-induced NASH (Okada et al., 2013). Similarly, atorvastatin had to be administered for 8 week to reduced Sirius red staining area in female Alms1 mutant ( foz/foz ) mice were fed a high-fat, high-sucrose diet containing 0.2% cholesterol for 24 weeks (Ioannou et al., 2015). We therefore anticipate that short duration of our therapy by atorvastatin for 3 weeks was reason, why we were unable to detect improvement in liver fibrosis. However, our intention was to describe initial steps in positive effect of atorvastatin in NASH, to identify primary target, which was successful.

Solution: We added the following statement to the discussion: Previous experimental studies also demonstrated improved liver fibrosis after statin administration (Chong et al., 2015; Park et al., 2016) which was not detected in our study. However, in a corresponding high-fat high-cholesterol diet-induced model of steatosis, 8 weeks of atorvastatin administration were necessary to reduce liver fibrosis. Therefore, we anticipate that 3 weeks of atorvastatin’s administration that identified primary steps of its hepatoprotective mechanisms in NASH were too short to develop an antifibrotic effect.   

  1. Please discuss why the cholesterol was not changed by Atorvastatin (line 463-464). The dosage of admiration Atorvastatin is proper? Please described it

Unchanged plasma cholesterol concentration were seen in crucial work evaluating effect of statins on high fat diet induced NAFLD using mice models (Van Rooyen et al., 2013; Orime et al., 2016; Park et al., 2016) while others seen reduction (He et al., 2017; Zhang et al., 2018). This result may be explained by the fact that among all plasma lipoproteins, statins specifically target the levels of LDL-cholesterol, which, compared to humans, is underrepresented in mice (Yin et al., 2012; Caparros-Martin et al., 2017). It is also well known that statins strongly induce expression of Hmgcr via a SREBP2-mediated pathway (Roglans et al., 2002). Measurements of organ-specific cholesterol synthesis revealed that hepatic cholesterol synthesis was profoundly increased in statin-treated animals following the extent of overexpression of the enzyme (Schonewille et al., 2016). As a consequence, trough plasma concentrations of statins before the next dose may lead to a short relief of inhibition leading to an overshoot of synthesis. An overshoot of cholesterol synthesis upon statin withdrawal has been shown in rats (Bilhartz et al., 1989; Fujioka et al., 1995). These results are in accordance with the study of Chuang et al. (Chuang et al., 2014) who have measured an increase in acute cholesterol synthesis in BALB/c mice treated with simvastatin. It is therefore possible that brief increase in cholesterol synthesis in between inhibition by peak concentration may also contribute to unchanged plasma cholesterol concentrations. Third mechanism responsible for the maintenance of cholesterol concentrations during statin therapy might be the reduction of biliary cholesterol secretion and saturation index which has been observed in humans treated with pravastatin (Reihnér et al., 1990; Kallien et al., 1999) or lovastatin (Hanson and Duane, 1994). In addition, statin use was associated with a reduction in the risk of gallstone disease (Bodmer et al., 2009), supporting decreased rather than increased biliary cholesterol secretion rates.

Importantly, improving NASH by statins is independent of their cholesterol reducing effect and rather corresponds with their pleiotropic effect (Park et al., 2016; Schierwagen et al., 2016), which is consistent with reduced inflammation in our study.

We used a dose of atorvastatin (20 mg/kg/day) that is relatively conservative for mice (others used 10, 15, 20 or 30 mg/kg/day (Rodríguez-Calvo et al., 2009; Matafome et al., 2011; Vila et al., 2011; Van Rooyen et al., 2013; Ioannou et al., 2015; Park et al., 2016), noting that hepatic drug clearance is ~ »10-fold higher in rodents than in humans.

Solution: We extended the list of relevant publications referring unchanged cholesterol concentration during statin therapy in NASH and added following sentence to discussion This result may be explained by the fact that statins primarily control levels of LDL cholesterol, which, compared to humans, is a minor component in plasma lipoproteins of mice (Schonewille et al., 2016).

  1. If possible, please show the changes in microbiota by atorvastatin administration.

Thank you for this suggestion. Honestly, we have prepared such an analysis with our Microbiology department in our University Hospital more than one year ago, but then COVID epidemy started and the department provides PCR and sequencing COVID diagnostics for East Bohemia region, thus all services for preclinical research were canceled.

  1. In figure 1, the characters of “C” “CA” “F”, “FA” was overlap by the image of liver histology

I am sorry for this mistake. We have corrected it. The whole figure was embedded as a single file.

References

Athyros VG, Alexandrides TK, Bilianou H, Cholongitas E, Doumas M, Ganotakis ES, Goudevenos J, Elisaf MS, Germanidis G, Giouleme O, Karagiannis A, Karvounis C, Katsiki N, Kotsis V, Kountouras J, Liberopoulos E, Pitsavos C, Polyzos S, Rallidis LS, Richter D, Tsapas AG, Tselepis AD, Tsioufis K, Tziomalos K, Tzotzas T, Vasiliadis TG, Vlachopoulos C, Mikhailidis DP and Mantzoros C (2017) The use of statins alone, or in combination with pioglitazone and other drugs, for the treatment of non-alcoholic fatty liver disease/non-alcoholic steatohepatitis and related cardiovascular risk. An Expert Panel Statement. Metabolism 71:17-32.

Athyros VG, Mikhailidis DP, Didangelos TP, Giouleme OI, Liberopoulos EN, Karagiannis A, Kakafika AI, Tziomalos K, Burroughs AK and Elisaf MS (2006) Effect of multifactorial treatment on non-alcoholic fatty liver disease in metabolic syndrome: a randomised study. Curr Med Res Opin 22:873-883.

Bilhartz LE, Spady DK and Dietschy JM (1989) Inappropriate hepatic cholesterol synthesis expands the cellular pool of sterol available for recruitment by bile acids in the rat. J Clin Invest 84:1181-1187.

Bodmer M, Brauchli YB, Krähenbühl S, Jick SS and Meier CR (2009) Statin use and risk of gallstone disease followed by cholecystectomy. Jama 302:2001-2007.

Caparros-Martin JA, Lareu RR, Ramsay JP, Peplies J, Reen FJ, Headlam HA, Ward NC, Croft KD, Newsholme P, Hughes JD and O'Gara F (2017) Statin therapy causes gut dysbiosis in mice through a PXR-dependent mechanism. Microbiome 5:95.

Dongiovanni P, Petta S, Mannisto V, Mancina RM, Pipitone R, Karja V, Maggioni M, Kakela P, Wiklund O, Mozzi E, Grimaudo S, Kaminska D, Rametta R, Craxi A, Fargion S, Nobili V, Romeo S, Pihlajamaki J and Valenti L (2015) Statin use and non-alcoholic steatohepatitis in at risk individuals. J Hepatol 63:705-712.

Doumas M, Imprialos K, Dimakopoulou A, Stavropoulos K, Binas A and Athyros VG (2018) The Role of Statins in the Management of Nonalcoholic Fatty Liver Disease. Curr Pharm Des 24:4587-4592.

Fujioka T, Nara F, Tsujita Y, Fukushige J, Fukami M and Kuroda M (1995) The mechanism of lack of hypocholesterolemic effects of pravastatin sodium, a 3-hydroxy-3-methylglutaryl coenzyme A reductase inhibitor, in rats. Biochim Biophys Acta 1254:7-12.

Hanson DS and Duane WC (1994) Effects of lovastatin and chenodiol on bile acid synthesis, bile lipid composition, and biliary lipid secretion in healthy human subjects. J Lipid Res 35:1462-1468.

He X, Zheng N, He J, Liu C, Feng J, Jia W and Li H (2017) Gut Microbiota Modulation Attenuated the Hypolipidemic Effect of Simvastatin in High-Fat/Cholesterol-Diet Fed Mice. J Proteome Res 16:1900-1910.

Hyogo H, Yamagishi S, Maeda S, Kimura Y, Ishitobi T and Chayama K (2012) Atorvastatin improves disease activity of nonalcoholic steatohepatitis partly through its tumour necrosis factor-α-lowering property. Digestive and liver disease : official journal of the Italian Society of Gastroenterology and the Italian Association for the Study of the Liver 44:492-496.

Chong LW, Hsu YC, Lee TF, Lin Y, Chiu YT, Yang KC, Wu JC and Huang YT (2015) Fluvastatin attenuates hepatic steatosis-induced fibrogenesis in rats through inhibiting paracrine effect of hepatocyte on hepatic stellate cells. BMC Gastroenterol 15:22.

Chuang JC, Valasek MA, Lopez AM, Posey KS, Repa JJ and Turley SD (2014) Sustained and selective suppression of intestinal cholesterol synthesis by Ro 48-8071, an inhibitor of 2,3-oxidosqualene:lanosterol cyclase, in the BALB/c mouse. Biochem Pharmacol 88:351-363.

Ioannou GN, Van Rooyen DM, Savard C, Haigh WG, Yeh MM, Teoh NC and Farrell GC (2015) Cholesterol-lowering drugs cause dissolution of cholesterol crystals and disperse Kupffer cell crown-like structures during resolution of NASH. J Lipid Res 56:277-285.

Kallien G, Lange K, Stange EF and Scheibner J (1999) The pravastatin-induced decrease of biliary cholesterol secretion is not directly related to an inhibition of cholesterol synthesis in humans. Hepatology 30:14-20.

Matafome P, Louro T, Rodrigues L, Crisostomo J, Nunes E, Amaral C, Monteiro P, Cipriano A and Seica R (2011) Metformin and atorvastatin combination further protect the liver in type 2 diabetes with hyperlipidaemia. Diabetes Metab Res Rev 27:54-62.

Nelson A, Torres DM, Morgan AE, Fincke C and Harrison SA (2009) A pilot study using simvastatin in the treatment of nonalcoholic steatohepatitis: A randomized placebo-controlled trial. J Clin Gastroenterol 43:990-994.

Okada Y, Yamaguchi K, Nakajima T, Nishikawa T, Jo M, Mitsumoto Y, Kimura H, Nishimura T, Tochiki N, Yasui K, Mitsuyoshi H, Minami M, Kagawa K, Okanoue T and Itoh Y (2013) Rosuvastatin ameliorates high-fat and high-cholesterol diet-induced nonalcoholic steatohepatitis in rats. Liver Int 33:301-311.

Orime K, Shirakawa J, Togashi Y, Tajima K, Inoue H, Nagashima Y and Terauchi Y (2016) Lipid-lowering agents inhibit hepatic steatosis in a non-alcoholic steatohepatitis-derived hepatocellular carcinoma mouse model. Eur J Pharmacol 772:22-32.

Park HS, Jang JE, Ko MS, Woo SH, Kim BJ, Kim HS, Park HS, Park IS, Koh EH and Lee KU (2016) Statins Increase Mitochondrial and Peroxisomal Fatty Acid Oxidation in the Liver and Prevent Non-Alcoholic Steatohepatitis in Mice. Diabetes Metab J 40:376-385.

Reihnér E, Rudling M, Ståhlberg D, Berglund L, Ewerth S, Björkhem I, Einarsson K and Angelin B (1990) Influence of pravastatin, a specific inhibitor of HMG-CoA reductase, on hepatic metabolism of cholesterol. N Engl J Med 323:224-228.

Rodríguez-Calvo R, Barroso E, Serrano L, Coll T, Sánchez RM, Merlos M, Palomer X, Laguna JC and Vázquez-Carrera M (2009) Atorvastatin prevents carbohydrate response element binding protein activation in the fructose-fed rat by activating protein kinase A. Hepatology 49:106-115.

Roglans N, Verd JC, Peris C, Alegret M, Vázquez M, Adzet T, Díaz C, Hernández G, Laguna JC and Sánchez RM (2002) High doses of atorvastatin and simvastatin induce key enzymes involved in VLDL production. Lipids 37:445-454.

Sarkar S, Terry JG, Ikizler TA, Crouse JR, 3rd, Carr JJ and Hung AM (2016) The effect of high intensity statin use on liver density: A post hoc analysis of the coronary artery calcification treatment with zocor [CATZ] study. Obesity research & clinical practice 10:613-615.

Schierwagen R, Maybüchen L, Hittatiya K, Klein S, Uschner FE, Braga TT, Franklin BS, Nickenig G, Strassburg CP, Plat J, Sauerbruch T, Latz E, Lütjohann D, Zimmer S and Trebicka J (2016) Statins improve NASH via inhibition of RhoA and Ras. Am J Physiol Gastrointest Liver Physiol 311:G724-g733.

Schonewille M, de Boer JF, Mele L, Wolters H, Bloks VW, Wolters JC, Kuivenhoven JA, Tietge UJF, Brufau G and Groen AK (2016) Statins increase hepatic cholesterol synthesis and stimulate fecal cholesterol elimination in mice. Journal of lipid research 57:1455-1464.

Van Rooyen DM, Gan LT, Yeh MM, Haigh WG, Larter CZ, Ioannou G, Teoh NC and Farrell GC (2013) Pharmacological cholesterol lowering reverses fibrotic NASH in obese, diabetic mice with metabolic syndrome. J Hepatol 59:144-152.

Vila L, Rebollo A, Adalsteisson GS, Alegret M, Merlos M, Roglans N and Laguna JC (2011) Reduction of liver fructokinase expression and improved hepatic inflammation and metabolism in liquid fructose-fed rats after atorvastatin treatment. Toxicol Appl Pharmacol 251:32-40.

Yin W, Carballo-Jane E, McLaren DG, Mendoza VH, Gagen K, Geoghagen NS, McNamara LA, Gorski JN, Eiermann GJ, Petrov A, Wolff M, Tong X, Wilsie LC, Akiyama TE, Chen J, Thankappan A, Xue J, Ping X, Andrews G, Wickham LA, Gai CL, Trinh T, Kulick AA, Donnelly MJ, Voronin GO, Rosa R, Cumiskey AM, Bekkari K, Mitnaul LJ, Puig O, Chen F, Raubertas R, Wong PH, Hansen BC, Koblan KS, Roddy TP, Hubbard BK and Strack AM (2012) Plasma lipid profiling across species for the identification of optimal animal models of human dyslipidemia. J Lipid Res 53:51-65.

Zhang W, Yang X, Chen Y, Hu W, Liu L, Zhang X, Liu M, Sun L, Liu Y, Yu M, Li X, Li L, Zhu Y, Miao QR, Han J and Duan Y (2018) Activation of hepatic Nogo-B receptor expression-A new anti-liver steatosis mechanism of statins. Biochim Biophys Acta Mol Cell Biol Lipids 1863:177-190.

Round 2

Reviewer 2 Report

The authors replied to my comments properly.